# Self-supervision through Random Segments with Autoregressive Coding (RandSAC)

**Tianyu Hua**[1,4,6]  **Yonglong Tian**[2]  **Sucheng Ren**[3]  **Michalis Raptis**[4]  **Hang Zhao**[5]  **Leonid Sigal**[1,6,7]
[1]University of British Columbia        [2]Massachusetts Institute of Technology
[3]South China University of Technology        [4]Google Research
[5]Tsinghua University        [6]Vector Institute for AI        [7]Canada CIFAR AI Chair

## Abstract

Inspired by the success of self-supervised autoregressive representation learning in natural language (GPT and its variants), and advances in recent visual architecture design with Vision Transformers (ViTs), in this paper, we explore the effect various design choices have on the success of applying such training strategies for *visual* feature learning. Specifically, we introduce a novel strategy that we call **Rand**om **S**egments with **A**utoregressive **C**oding (RandSAC). In RandSAC, we group patch representations (image tokens) into hierarchically arranged segments; within each segment, tokens are predicted in parallel, similar to BERT, while across segment predictions are sequential, similar to GPT. We illustrate that randomized serialization of the segments significantly improves the performance and results in distribution over spatially-long (across-segments) and -short (within-segment) predictions which are effective for feature learning. We illustrate the pertinence of these design choices and explore alternatives on a number of datasets (*e.g.*, CIFAR10, CIFAR100, ImageNet). While our pre-training strategy works with vanilla Transformer, we also propose a conceptually simple, but highly effective, addition to the decoder that allows learnable skip-connections to encoder's feature layers, which further improves the performance.

## 1 Introduction

Deep learning has powered enormous successes in Computer Vision and NLP over the past 10, or so, years. It has lead to significant improvements in object detection (Redmon et al., 2016), segmentation (He et al., 2017), as well as higher-level cognition tasks (*e.g.*, Visual Question Answering (Antol et al., 2015), Visual Navigation (Mayo et al., 2021), *etc.*). These successes have been enabled by both advances in parallel hardware (GPUs) and, perhaps more importantly, large-scale task-specific labeled datasets that allow supervised learning. This appetite for large data has, until very recently, stagnated progress, particularly in building general-purpose visual architectures.

These types of considerations date back to the early days of machine learning, and deep learning in particular, where it has long been postulated that unsupervised, or self-supervised, learning could allow learning of robust and general feature representations that can then be readily used (or fine-tuned) to target tasks. Self-supervised learning has been explored in computer vision in various forms: denoising autoencoders (Pathak et al., 2016; Vincent et al., 2008), colorization (Zhang et al., 2016) or jigsaw puzzle (Doersch et al., 2015; Noroozi & Favaro, 2016) proxy objectives. However, the success of such self-supervised pre-training was somewhat limited. In contrast, the success of similar self-supervised ideas in NLP has been much more dominant with GPT (Brown et al., 2020) and BERT (Devlin et al., 2018) architectures, and their variants. These pre-training strategies now enable state-of-the-art performance on a wide array of natural language tasks.

Recent advances in vision architectures, such as Vision Transformers (ViT) (Dosovitskiy et al., 2021; Liu et al., 2021), which serialize visual 2d data, have opened an opportunity to apply similar large scale pre-training techniques in vision, with increasing successes. Self-supervised pre-training techniques with ViTs can be characterized into two broad categories: *contrastive* and *predictive*; as well as their combinations. In *contrastive* learning, pre-training architectures are learned to be invariant to certain perturbations in data (*e.g.*, spatial shifts, color jitter) by forming positive and

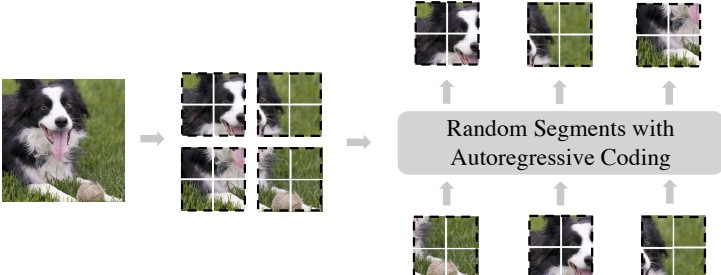

Figure 1: **Randomized Autoregressive Segment Prediction.** Illustration of our autoregressive segment prediction framework (RandSAC). RandSAC breaks the image into tokens which are arranged into segments (here squares of size $2 \times 2$). The autoregressive (GPT-style) transformer-based model is then trained to predict segments in a randomly sampled serialization order. As a result, tokens within segments are predicted in parallel, while segments themselves are predicted sequentially.

negative pairings of augmented data samples. This is a powerful technique, but requires designers to make assumptions about invariances that the architecture should learn. In addition, purely contrastive models tend to incorporate *center bias* (Chen et al., 2022; 2021a), which makes them less transferable for tasks such as segmentation where non-object centric regions need to be modeled. Alternatively, *predictive* models learn to predict elements of the scene, either in parallel by reconstructing masked regions/tokens (Bao et al., 2022; He et al., 2021) (*a.k.a.*, masked image modeling or BERT-style pre-training) or to predict images in auto-regressive language-modeling manner (Chen et al., 2020a) (*a.k.a.*, GPT-style pre-training). It is interesting to observe that on the NLP side, GPT models have shown to be powerful, while vision models have gravitated more towards BERT-style pre-training both with visual (Chen et al., 2020a; Bao et al., 2022) and multi-modal data (Lu et al., 2019; Su et al., 2020).

Motivated by this, we adopt an autoregressive pre-training strategy (see Figure 1) and ask a number of important empirical questions about the use of such pre-training and what makes it effective. Specifically, (1) we ask what granularity (scale) and shape of tokens (patches, blobs) is most effective and how it affects the performance? (2) How best to serialize predictions? For example, previous approaches, such as image GPT (Chen et al., 2020a), leveraged raster ordering. While such ordering is perhaps "optimal" from correlation and predictive/generative (van den Oord et al., 2016) points of view, we show that it is not optimal for general feature learning. We also explore (3) whether deterministic vs. stochastic tokenization and serialization are helpful. Finally, (4) we explore the effective interactions between the decoder and encoder layers; proposing a new ViT architecture that uses learned skip connections between encoder and decoder layers to improve performance.

**Contributions.** We make two core contributions. First, we propose a new pre-training strategy that leverages (randomly) sampled hierarchical segment cluster traversals to autoregresively train ViT models. This allows both short- and long-term spatial predictions, allowing distribution over easy and hard predictive tasks[1]. We note that the effectiveness of single random segment inpainting was initially observed in (Pathak et al., 2016), but is notably missing from most recent self-supervised strategies. Our pre-training strategy generalizes this observation and strategy to hierarchical and serialized predictions. Second, we propose a flexible ViT decoder that at each decoding layer learns to dynamically attend over different levels of features in the encoder. This in effect creates learned skip-connections, as compared to UNet (Ronneberger et al., 2015) and others that require fixed connections in a symmetric encoder-decoder design, which further improve the performance.

**Discussion.** The above pre-training strategy, while empirically motivated, is also loosely modeled after human vision. Humans attend to the scene by a sequence of foveal observations, where an eye shifts over a series of fixation points; such motions are called *saccades*. Some saccades are long-range and voluntary, while others are local and involuntary (*a.k.a.*, microsaccades (Rolfs, 2009)). Our segments can be "viewed" as predictive foveal regions, and the hierarchical serialization of such regions as the combination of micro and macro saccades. The significant difference from human vision, is that in human vision saccades are purposeful and have been shown to be conditioned on the task (Yarbus, 1967). In contrast, our pre-training such "saccadic" movements are randomly sampled.

---

[1]This is, in part, motivated by (He et al., 2021) which observe that in BERT-style pre-training high amount of masking (as much as 75%), which corresponds to harder predictive tasks, leads to better feature learning.

Learning a purposeful policy for hierarchical serialization of segments, would be an interesting future work. However, this is a difficult task that is beyond the scope of this paper.

## 2   RELATED WORK

**Transformer-based Natural Language Modeling.**   In the field of natural language processing (NLP), two dominant self-supervised language modeling paradigms are Masked Language Modeling, such as BERT (Devlin et al., 2018), and GPT-style autoregressive pre-training (Brown et al., 2020; Radford & Narasimhan, 2018; Radford et al., 2019). Given a sentence, BERT and its variants (Lan et al., 2020; Liu et al., 2019) pre-train transformer encoders by predicting randomly masked out input words, referred to as *tokens*. Such frameworks model the bidirectional (contextual) dependencies between the visible tokens and the corrupted/masked tokens. GPT, which can be viewed as a special case of the transformer decoder, on the other hand, models the left-to-right natural order of languages. Recent advances in large-scale generative language modeling show powerful few-shot capabilities and are believed to be a promising path towards general machine intelligence. Permutation-based autoregressive model (Yang et al., 2019) was proposed to bridge the gap between autoregressive language modeling and masked autoencoding by maximizing the likelihood over all permutations of the factorization order. We take inspiration from GPT-style autoregressive pre-training in formulating our model, and focus on important aspects of mapping such strategy onto visual (ViT) models, where tokenization and serialization are not as well defined as in language.

**Contrastive Image Learning.**   Contrastive methods (Chen et al., 2020b; He et al., 2020; van den Oord et al., 2018; Tian et al., 2020) and their negative-sample-free variants (Chen & He, 2021; Grill et al., 2020; Hua et al., 2021; Zbontar et al., 2021) have emerged as a dominant research direction for unsupervised/self-supervised visual representation learning over the past 1–2 years. By building agreement among augmented versions of the input data, image features that are invariant of those perturbations can be learned. This method implicitly assumes a set of representational invariance (*e.g.*, color and spatial invariance). Once such representations are learned they are either used directly, or fine-tuned, to one or more downstream supervised tasks (*e.g.*, classification, detection, segmentation). When a downstream task violates the aforementioned invariance assumptions, they display poor transferability (Xiao et al., 2021). For example, the center-bias (Chen et al., 2022) and small-object feature suppression (Chen et al., 2021a) have been observed in prior works. Masked image modeling & autoregressive image encoding, of which our method is an instance, tend to perform better in such circumstances (Bao et al., 2022; He et al., 2021).

**Masked Image Modeling.**   Early CNN-based masked image modeling, also known as image inpainting (Doersch et al., 2015; Pathak et al., 2016; Yu et al., 2018), has shown promising results but failed to become a predominant training paradigm, in part, due to its inferior performance with respect to large-scale supervised pre-training (*e.g.*, on ImageNet). The recent trend of incorporating transformers into vision architectures (Carion et al., 2020), or replacing CNN completely (Dosovitskiy et al., 2021), by tokenizing images into a grid of non-overlapping patches, have enabled application of large scale NLP pretraining techniques in vision, *e.g.*, (Bao et al., 2022; He et al., 2021; Wei et al., 2021; Xie et al., 2022). Directly applying them to image pixels, however, leads to inferior performance (Chen et al., 2020a; Dosovitskiy et al., 2021). To this end, BEiT (Bao et al., 2022) proposes to predict discrete masked image tokens. Masked Autoencoder (MAE) (He et al., 2021) suggests a 75% random masking ratio for image modeling; and SimMIM (Xie et al., 2022) studies different masking strategies for pretraining. MaskFeat (Wei et al., 2021) investigates five different reconstruction targets. TinyMIM (Ren et al., 2023) introduces distilling token relations from MAE model, and SplitMask (El-Nouby et al., 2021) illustrates the ability of BEiT to train with small scale pre-training datasets. Our proposed RandSAC strategy, is related to masked image modeling, but is autoregressive in nature.

**Autoregressive Image Encoding.**   Compared with BERT-style pre-training for vision transformers, GPT-like autoregressive models have been overlooked due to their complexity introduced by dense image pixels. In image GPT (Chen et al., 2020a), images are limited to $64 \times 64 = 4096$ pixels. The 4096 pixels are tokenized and serialized in raster-order before feeding into a causal transformer. The quadratic time/space complexity of self-attention prevents the scaling of such approaches.

## 3   RANDOM SEGMENT WITH AUTOREGRESSIVE CODING

RandSAC learns representations through autoregressive image segment prediction. It partitions a tokenized image into random spatially coherent non-overlapping (hierarchical) segments, serializes them, and then autoregressively predicts tokens within these ordered segments. As a result, the token

predictions between segments are sequential, while within a segment are parallel. This training strategy has four important components that we will explore:

- ***Tokenization***. To use a transformer-based architecture, images need to be *tokenized*, *i.e.*, transformed into a set of basic image elements. For example, some approaches discretize images (Chen et al., 2020a), while others patchify them (Cordonnier et al., 2020; Bao et al., 2022; Dosovitskiy et al., 2021; He et al., 2021; Xie et al., 2022). Tokenization strategy dictates the scale and number of tokens, which affects performance and computation cost.

- ***Segment Partitioning***. After tokenizing the image, the tokens are grouped into spatially coherent segments. Those segments are autoregressively predicted following some prescribed serialization order. The size and shape of segments and the way they are traversed can affect training and downstream performance.

- ***Serialization Strategy***. Serialization strategy affects the traversal order of segments. In prior autoregressive modeling (Chen et al., 2020a) raster-order is assumed. We show that stochastic (*i.e.*, randomized) serialization is much more effective.

- ***Transformer Architecture***. In a GPT-style autoregressive model, the target sequence is identical to the shifted input sequence throughout training. However, for random segment prediction, the target sequence order varies for each sample. To enable this, we leverage a transformer decoder which takes as input position of each token and outputs its predicted representation conditioned on the transformer encoded context. In addition, we propose a novel trainable skip-connection layer for efficient decoding.

In the following section, the default option for model architecture is the vanilla masked transformer introduced in Section 4. We experiment with two different datasets, CIFAR10 (Krizhevsky, 2009) and, where appropriate, ImageNet100 (Tian et al., 2020). Evaluation protocols are described in Section 5, and implementation details are in the Supplemental. We use a simple mean square error (MSE) as our pixel reconstruction objective.

## 3.1 FROM PIXELS TO TOKENS

**Tokenization.** We start from raster-order serialization and compare two different tokenization strategies introduced by iGPT (Chen et al., 2020a) and ViT (Dosovitskiy et al., 2021). Assume a dataset $\mathcal{D}$ of images $\mathbf{X} \in \mathbb{R}^{H \times W \times C}$, where $H, W, C$ are the height, width, and the number of channels of the image. We reshape each image into $N = HW/P^2$ patches, where $P$ is the resolution of each patch. Tokens are obtained by linearly projecting the patches $\mathbf{X} = \{\mathbf{x}_i\}_{i=1}^N$ and serialized row-by-row.

For pixel prediction experiment, we set $P = 1$, letting image patch size be $1 \times 1$ pixels (see Figure 2 (b)). For ViT style patch prediction experiment, we split the $32 \times 32$ CIFAR10 image into $8 \times 8 = 64$ patches (see Figure 2 (c)), each patch consists of $4 \times 4$ pixels ($P = 4$). Note that for a fair comparison, we didn't strictly follow iGPT, where they minimize the negative log-likelihood of the quantized RGB values. We simply adopt a mean squared error (MSE) between the predicted and target pixel values for all our experiments following (He et al., 2021). Note that for visualizations in Figure 2 we use a downsampled CIFAR10 image.

The results for these two tokenization options are illustrated in Table 1 (additional scales are in Supplemental) under `pixel-raster` and `patch-raster` respectively in terms of linear probing and fine-tuning accuracy (see Sec. 5.1 for definition of metrics). From the

|            | pixel-raster | patch-raster |
|------------|:------------:|:------------:|
| **LIN**(↑) | 41.70        | **55.53**    |
| **FT**(↑)  | 59.35        | **78.67**    |

Table 1: **Tokenization** on CIFAR10.

point of view of representation learning, patches are substantially better. Further, computationally, the self-attention mechanism in a transformer uses $O(n^2)$ in both time and space with respect to the sequence length. Hence for pixel tokenization, the complexity is $O((HW)^2)$. For patches, the complexity is reduced to $O((HW/P^2)^2)$. In our CIFAR10 experiment, when $P = 4$, the complexity of training is lowered by a factor of $P^4 = 256$. Hence, **patches result in better tokenization**.

**Stochastic Serialization.** Randomized pretext tasks play an important role in a range of self-supervised learning algorithms. In NLP, for example, (Yang et al., 2019) improves fixed-order autoregressive language models by allowing all possible permutations of the factorization order

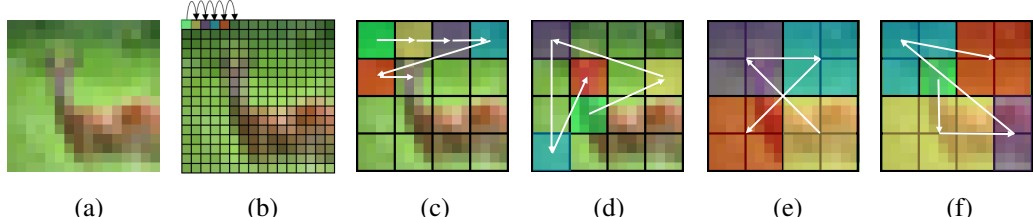

| | | | | | |
|---|---|---|---|---|---|
| (a) | (b) | (c) | (d) | (e) | (f) |

Figure 2: **Autoregressive Prediction Schemes.** Left-to-right: (a) original image from CIFAR 10; (b) raster-order pixel prediction; (c) raster-order patch prediction; (d) stochastic patch prediction; (e) stochastic square segment prediction ($M = 2$); (f) stochastic blob segment prediction ($K = 5$).

during training. For autoregressive ViT training of stochastic token serialization, we adopt a similar strategy by shuffling the token sequence for each image sample. Note that this does *not* mean that our prediction sequence is "orderless". By moving from fixed raster-order prediction to randomized sequence prediction, keeping all else the same, we observe 20% improvement in linear evaluation and ~10% in

|  | patch-raster | patch-random |
|---|---|---|
| **CF10-LIN**($\uparrow$) | 55.53 | **75.53** |
| **CF10-FT**($\uparrow$) | 78.67 | **87.52** |
| **IN100-LIN**($\uparrow$) | 49.35 | **53.02** |
| **IN100-FT**($\uparrow$) | 82.13 | **84.15** |

Table 2: **Serialization** on CIFAR10 and ImageNet100.

fine-tuning (Table 2 CIFAR10). Improvements on ImageNet100 are more modest (3.67% and ~2% respectively), but still significant and overall **stochastic serialization is clearly superior**.

### 3.2 GROUPING TOKENS INTO SEGMENTS

In this section, we introduce a concept of *segments*, which we define as groups (or clusters) of tokens. Effectively each segment forms an equivalency class within our serialized order, where tokens are encoded and decoded in parallel. Across segments, however, predictions are still strictly sequential. The motivation for introducing segments is two-fold. First, it allows us to reduce the overall number of autoregressive prediction steps. Second, it allows our autoregressive strategy to effectively leverage aspects of parallel, BERT-style, prediction locally. The autoregressive prediction steps can also be changed without introducing parallel prediction, simply by changing the patch size $P$. This is ineffective, however, as we show in Supplemental Section A.1. In what follows, we experiment with two spatially coherent segment strategies (square and blob) and then look at the importance of this spatial coherence in segment formation.

**Square Segments.** Once we have a grid of $N$ patches of size $\frac{H}{P} \times \frac{W}{P}$, we reshape the tokens into a set of square segments $M \times M$, where the $M$ denotes the size of the square. The segment count $K$ of an image of $H \times W$ is thus defined by: $K = \frac{H \times W}{(P \times M)^2}$. For example, in our CIFAR10 experiment, an input image of size $32 \times 32$ is tokenized into a grid of $8 \times 8$ tokens, each of which is a $4 \times 4$ pixel patch. We set the square size $M = 2$. The tokens are then split into $(8/2)^2 = 16$ segments, which are shuffled

| Square size M | 1 | 2 | 4 |
|---|---|---|---|
| **LIN**($\uparrow$) | 75.53 | **81.38** | 79.38 |
| **FT**($\uparrow$) | 87.52 | **91.38** | 90.23 |

Table 3: **Square-random** serialization as a function of $M$ on CIFAR10.

randomly for autoregressive prediction as before. We list the representation quality with different square segment size ($M$) in Table 3. Since the grid size is $8 \times 8$ for CIFAR10, we chose square sizes $M = [1, 2, 4]$. Note that, when $M = 8$, there will be only one segment (*e.g.*, $K = 1$) and no prediction can be made; $M = 1$ is equivalent to no segments (*i.e.*, patch-random in Table 2).

**Blob Segments.** We define blob segments as irregular elliptical segments defined by a sampled Mixture of Gaussians. To obtain $K$ random blobs for a given image, we first sample $K$ Gaussians with a range of means and standard deviations in the image space. Then we simply assign each token $\mathbf{x}_i$ which is at position $(x_i, y_i)$ to the closest mixture component using Mahalanobis distance. We illustrate the square and blob strategies in Figure 2 (e) and (f), respectively. Note that beyond the shape, blob segments allow for variability in size squares do not. See details in Suppl. Section A.2.

**Analysis.** As can be seen from Table 4, both square segments and blob segments surpass segment-

free patch-based autoregression (see `square-random` and `blob-random` compared with `patch-random`). The blob segments and square segments behave similarly. In addition, with blobs, we can easily modify the number of segments. However, with squares, the

| | patch-random | square-random | blob-random |
|---|---|---|---|
| **CF10-LIN**($\uparrow$) | 75.53 | **81.38** | **82.52** |
| **CF10-FT**($\uparrow$) | 87.52 | **91.38** | **91.53** |
| **IN100-LIN**($\uparrow$) | 53.02 | **64.78** | **65.00** |
| **IN100-FT**($\uparrow$) | 84.15 | **86.22** | **85.16** |

Table 4: **Segments** on CIFAR10.

segment number is constrained by the token number. A grid of $8 \times 8$ tokens can either be segmented into $4 \times 4$ or $2 \times 2$ squares. A grid size of $13 \times 13$ can not be divided into any kind of squares. Blob segments, on the other hand, are more flexible.

**Do segments need to be spatially coherent?** The idea of a "segment" puts emphasis on the spatial coherence of the tokens. The upper part of Table 5 shows the performance of feature representations with respect to the number of blob segments $K$. In the bottom, we randomly shuffle all tokens so that tokens in any given "segment" no longer spatially coherent. We observe that feature learning

Table 5: **Segment Coherence.** Representation quality with different number of segments. Below we randomly permute the segments such that their spatial coherence is disrupted.

| Segment K | 3 | 5 | 7 | 9 | 11 |
|---|---|---|---|---|---|
| **LIN**($\uparrow$) | 80.87 | 81.82 | 82.52 | 81.88 | 82.02 |
| **FT**($\uparrow$) | 90.77 | 90.88 | 91.14 | 91.53 | 91.24 |
| **Shuffle** | 3 | 5 | 7 | 9 | 11 |
| **LIN**($\uparrow$) | 76.73 | 77.73 | 76.69 | 78.59 | 76.99 |
| **FT**($\uparrow$) | 89.63 | 89.75 | 89.22 | 90.00 | 89.15 |

deteriorates when segments are not spatially coherent. Note that segments without spatial coherence are still consistently better than `patch-random` from Table 4.

## 3.3 HIERARCHICAL SEGMENT SERIALIZATION

Images are hierarchical: a visual region of an image can often be interpreted as a component of a greater whole (Hinton, 2021) (*e.g.*, parts make up object, object scenes, and so on). Such compositionality motivates hierarchical groupings. In our case of random segment serialization, we postulate that similar hierarchical traversal order, which adds certain degree of locality, may be useful.

In Figure 3 we illustrate this concept that we operationalize. An image is first partitioned into 16 square segments, indicated by different colors and shades. We then group these 16 segments into 4 larger partitions following the same logic for segment generation. Different colors (*e.g.*, blue, orange, purple, and green) represent these partition groups; segments that share partition differ in shade. Hierarchical serialization is obtained by randomly, and sequentially, predicting the segments inside of each partition group (shown by the black arrows), and then jumping to another partition group at random. Note that the segment-level (local) and partition-level (global) serializations are both random. This idea can be extended to deeper hierarchies, with the depth of the hierarchy and grouping chosen based on the resolution and nature of the dataset.

Experimental results that compare flat serialization to two-level hierarchy are illustrated in Table 6. We perform these experiments on both CIFAR10 and ImageNet100 datasets. Our experiments show that **hierarchical serialization** and prediction consistently **outperform the flat counterparts**.

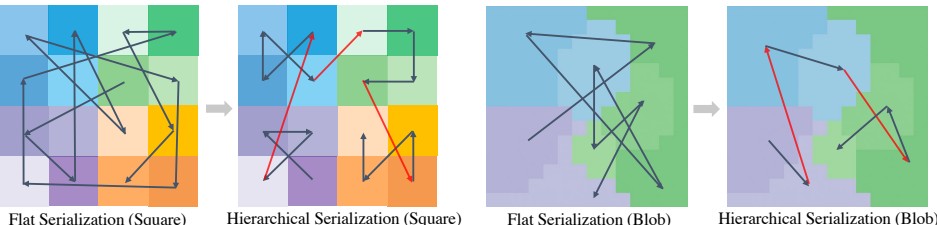

Flat Serialization (Square)  Hierarchical Serialization (Square)  Flat Serialization (Blob)  Hierarchical Serialization (Blob)

Figure 3: **Hierarchical Segment Serialization.** We partition an image into a hierarchy of segments (segments are illustrated by color and tokens within segment by shade). Autoregressive prediction is done by following a traversal of randomly generated hierarchical partitions.

Table 6: **Hierarchical Segment Prediction.** The number on top indicates the number of segments $K$ (*e.g.*, 4, 16) – flat/no-hierarchy; the $16 \rightarrow 4$ indicates hierarchical variants with two levels – 16 segments grouped into 4 partitions. Left/square and middle/blob results correspond to Fig. 3.

| Segments (square) | 4 | 16 | 16 →4 | Segments (Blob) | 3 | 7 | 7 →3 | 4 | 16 | 16 →4 |
|---|---|---|---|---|---|---|---|---|---|---|
| CF10 Linear (↑) | 79.38 | 81.38 | **82.46** | CF10 Linear (↑) | 80.87 | 82.52 | **82.71** | 81.09 | 80.97 | **82.61** |
| CF10 Fine-tune (↑) | 89.61 | 91.38 | **91.66** | CF10 Fine-tune (↑) | 90.77 | 91.14 | **91.20** | 90.63 | 90.57 | **91.15** |
| IN100 Linear (↑) | 55.88 | 64.90 | **65.81** | IN100 Linear (↑) | 56.26 | 63.36 | **64.64** | 60.62 | 64.50 | **64.92** |
| IN100 Fine-tune (↑) | 78.81 | 85.32 | **85.55** | IN100 Fine-tune (↑) | 81.34 | 84.36 | **84.48** | 83.07 | 86.06 | **86.18** |

## 4 ARCHITECTURE

Image GPT (Chen et al., 2020a) performs autoregressive prediction by shifting the source sequence one pixel to the right. Since the raster ordering of iGPT is fixed for all samples, the position for the next target token is implicitly modeled by the transformer. In contrast, in RandSAC, the next token depends on the serialization strategy, thus can vary from sample to sample during training. Moreover, when predicting the next segment, the tokens within each segment should be predicted jointly (in parallel). This requires lateral pathways that allow communication within target segments. To tackle the aforementioned problems, we propose to utilize the transformer decoder.

### 4.1 MASKED TRANSFORMER FOR SEGMENT PREDICTION

A standard transformer has an encoder-decoder structure (Vaswani et al., 2017). The encoder of a transformer maps a list of tokens $X = (\mathbf{x}_1, ..., \mathbf{x}_n)$ to a sequence of hidden representations $Z = (\mathbf{z}_1, ..., \mathbf{z}_n)$, also known as the *memory*. Given $X$ and source sequence $X_{src} = (\mathbf{x}_1, ..., \mathbf{x}_{n-1})$, during training, the decoder masks the internal attention matrix with a causal mask and predicts the target sequence $X_{tgt} = (\mathbf{x}_2, ..., \mathbf{x}_n)$ autoregressively. Each layer of the transformer encoder has two sub-layers: multi-head self-attention and a fully connected feed-forward network; both have residual connections. The decoder layer has a third attention sub-layer, which performs multi-head attention from the hidden representation $Z$ to the target representation $X_{tgt}$. We leverage attention masking to achieve autoregressive segment prediction using this framework; we discuss details next.

**Autoregressive Segment Encoder.** Figure 4 shows our transformer encoder block and a decoder block. We leave out the fully connected layer and residual connections for simplicity and only show the attentions. In this visualization, there are six patches. These six patches are then grouped into three segments denoted by colors: green, blue, and red. The random segment serialization order is green → blue → red. One layer of transformer encoder is illustrated on the left in light green. Serialized six patches/tokens with added fixed sine-cosine positional encoding are the input to the encoder. The encoder attention is masked following the serialized segment order: segments can attend to themselves and preceding segments only. They are restricted from looking at future segments using the, illustrated, *source mask*. Lastly, since the last segment does not have a succeeding segment, we only input the first four patches and leave out the two patches in the last segment.

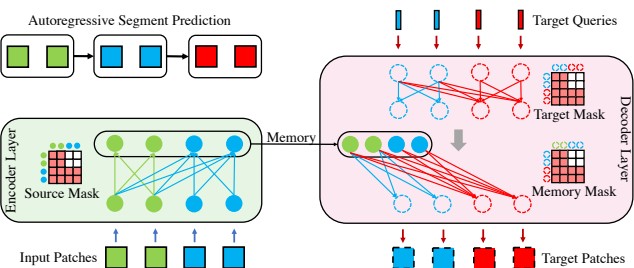

Figure 4: **Attention-masking for Autoregressive Segment Prediction.** For an image converted into a sequence of patches, we adopt a masked encoder-decoder transformer (Vaswani et al., 2017) for autoregressive segment prediction. In the encoder, causal *source mask* enables a given segment to only attend over preceding segments and the tokens within itself. The decoder, given the position of tokens (*i.e., target queries*), predicts tokens within each segment conditioned on encoded previous segments (enabled by the *memory mask*).

**Autoregressive Segment Decoder.** The input for the transformer decoder, illustrated on the right of Figure 4 in pink, is a set of fixed positional encodings to guide the reconstruction of target segments and tokens. Similar to the encoder input where we leave out the last segment patches, in the decoder, we shift the target sequence one segment to the left and ignore the positional encodings of the first segment because it does not have a preceding segment. The self-attention layer of the decoder is masked the same way as the encoder for autoregressive segment decoding. This layer enables co-attention over preceding and current segments for context.

**Evaluation.** During linear evaluation and fine-tuning, both the attention masks and decoder are removed, and the encoder is used as feature extractor for the downstream supervised classification.

## 4.2 TRAINABLE SKIP CONNECTIONS

The original transformer decoder layer can only attend to the same encoder output, often from the last layer of the encoder. In contrast, CNN encoder-decoder architectures are often symmetric with skip connections between encoder and decoder layers, *e.g.*, UNet (Ronneberger et al., 2015). We hypothesize that in our design, skip-connections between transformer encoder and decoder can similarly be beneficial. To enable such skip connections, we propose a trainable skip connection module that learns how to assign encoder memory to the decoder layers. Specifically, for a transformer with $L_{enc}$ and $L_{dec}$ number of layers, we learn a linear layer with parameters $\mathbf{W} \in \mathbb{R}^{L_{enc} \times L_{dec}}$, such that: $\mathbf{Z}^l = \sum_{k=1}^{L_{enc}} \mathbf{W}_{l,k} \mathbf{H}_{enc}^k$, where $\mathbf{H}_{enc}^k$ is an encoder representation from layer $k$ and $\mathbf{Z}^l$ is the formed memory for decoder layer $l$. Note, the linearly formed memory cells are conditioned on, and different, for each individual decoder layer. We refer the reader to the Supplemental Section A.3 for details and experiments that validate the effectiveness of this design and discuss efficiency.

## 5 EXPERIMENTS

We test RandSAC in two drastically different settings: low-data and ImageNet-1K pre-training. We evaluate the classification performance of our pretrained backbone with linear probing and fine-tuning. We also test the transfer of our ImageNet pretrained model (Suppl. Sec. B.1 and B.2).

**General Implementation Details.** We adopt minimal data augmentation strategy and use the *normalized pixel value* from (He et al., 2021) as our patch regression target. We obtain the reconstruction target by normalizing target pixels using the mean and standard deviation of the patch they belong. Our loss function computes the mean squared error (MSE) between the predicted pixel values and patch-normalized reconstruction target.

**Low-data Pretraining.** Vision transformers are known to be "data hungry" (Dosovitskiy et al., 2021) and require a large dataset and a series of data augmentations to pretrain (Touvron et al., 2021). To experiment in such a challenging setting, we evaluate our method on small-scale datasets. We train a "square" and "blob" RandSAC models using $16 \to 4$ and $11 \to 5$ hierarchies respectively.

**Pretraining on ImageNet-1K.** ImageNet ILSVRC-2012 (Deng et al., 2009) is a popular large scale image dataset with 1.28 million images and 1000 categories. We train "square" RandSAC ($16 \to 4$).

Detailed implementation details for all three settings are given in Supplemental Appendix D.

## 5.1 EVALUATION PROTOCOLS

**Linear Probing.** This measure is widely used for quantifying the quality of representation learning. It learns a linear classifier on top of the frozen feature of a pretrained encoder to classify the object-level classification labels. Then performance is evaluated using the val/test set.

**End-to-end Fine-tuning.** A recent study (Chen et al., 2022) shows that linear evaluation favors those methods with a *center-bias* such as contrastive learning. To complement linear probing, we also include 100-epoch fine-tuning evaluation. In fine-tuning, all parameters are optimized for classification. The fine-tuning recipe follows the common practice of supervised ViT training.

## 5.2 RESULTS

| | Model | Backbone | Parameter | Linear | Fine-tune |
|---|---|---|---|---|---|
| *Supervised* | DeiT (Touvron et al., 2021) | ViT-B | 86M | N/A | 81.2 |
| *Clustering* | DINO (Caron et al., 2021) | ViT-B | 86M | 78.2 | 82.8 |
| *Contrastive Learning* | MoCo v3 (Chen et al., 2021b) | ViT-B | 86M | 76.7 | 83.2 |
| *Masked Image Modeling* | BEIT (Bao et al., 2022) | ViT-B | 86M | N/A | 83.2 |
| | MAE (He et al., 2021) | ViT-B | 86M | 68.0 | 83.6 |
| *Autoregressive Image Modeling* | iGPT (Chen et al., 2020a) | iGPT-S | 76M | 41.9 | N/A |
| | iGPT (Chen et al., 2020a) | iGPT-M | 455M | 54.5 | N/A |
| | iGPT (Chen et al., 2020a) | iGPT-L | 1362M | 65.2 | N/A |
| | RandSAC-Square (K=9) | ViT-B | 86M | 72.3 | **83.7** |
| | RandSAC-Square (K=16→4) | ViT-B | 86M | 68.9 | **83.9** |

Table 8: **Comparison on ImageNet-1K**. Methods except for Autoregressive Image Modeling use image size $224 \times 224$. RandSAC uses image size 192 for pre-training and $224 \times 224$ for evaluation.

Table 7 shows low-data classification performance for clustering pre-training (DINO (Caron et al., 2021)), masked image encoding (MAE (He et al., 2021)) and our segment autoregressive coding (RandSAC). The MAE and DINO are pretrained using their official implementations. For MAE we use a 75% masking ratio as suggested in their paper. All models· are pretrained for 1600 epochs and evaluated with both 90-epoch linear probing (LIN) and 100-epoch fine-

| Model | CIFAR10 | | CIFAR100 | |
|---|---|---|---|---|
| | LIN | FT | LIN | FT |
| Supervised | | 91.3 | | 64.13 |
| DINO (Caron et al., 2021) | 89.0 | 94.4 | 65.78 | 76.3 |
| MAE (He et al., 2021) | 87.3 | 95.9 | 54.0 | 81.1 |
| RandSAC-Square | 92.1 | 96.7 | **69.7** | **81.5** |
| RandSAC-Blob | **93.9** | **96.9** | 67.9 | 79.6 |

Table 7: Low-data pre-training on CIFAR 10 and 100. RandSAC-Square uses $16 \rightarrow 4$ hierarchy while RandSAC-Blob uses $11 \rightarrow 5$.

tuning (FT). Under the low data benchmark, RandSAC outperforms other non-autoregressive algorithms and direct supervised training, by a large margin. Both the square and the blob hierarchical versions work well. We postulate that the superior performance of RandSAC comes from randomized segment prediction pretext task. The autoregressive coding objective that we propose, which is to traverse a hierarchy of randomly serialized visual segments, diversifies the small dataset, and serves as a sort of data augmentation.

Table 8 shows ImageNet pretraining result. We compare RandSAC with clustering (DINO (Caron et al., 2021)) and contrastive (MoCo v3 (Chen et al., 2021b)) transformer approaches, masked image encoding (BEIT (Bao et al., 2022) & MAE (He et al., 2021)), and our autoregressive counterpart iGPT (Chen et al., 2020a). We note, that due to limited access to computation, we were only able to run RandSAC once, without any parameter tuning. Nevertheless, RandSAC outperforms all predictive (non-contrastive methods) in linear probing, despite using a smaller image size for pretraining (192 vs 224). It is also among the best in fine-tuning (on par with MAE and better than the rest).

Contrastive models do tend to perform better in linear probing, but also differ in pre-training. For example, contrastive methods require two global crops of the input image while other methods only process one crop; DINO uses **10** local crops. In addition, linear probing for DINO and iGPT is evaluated using the last 4 and 5 transformer blocks, respectively, while MoCo v3, MAE, and RandSAC only evaluate the last block output. A longer feature vector tends to result in better linear probing accuracy (Caron et al., 2021; Chen et al., 2020a). Lastly, it is worth mentioning that RandSAC can be easily combined with contrastive objectives in the future.

## 6 CONCLUSION

We present a new self-supervised pre-training strategy we call RandSAC. In doing so, we also study and provide general insights into ViT pre-training (*e.g.*, tokenization, segmentation, and serialization). We found randomized serialization of hierarchical image segments significantly improves autoregressive pre-training of ViTs. In addition, we propose a new design for the transformer decoder, which facilitates improved performance. We show evidence that the proposed task and model could be the key to developing a powerful GPT-like model for visual representation learning.

ACKNOWLEDGMENTS

This work was funded, in part, by the Vector Institute for AI, Canada CIFAR AI Chair, NSERC CRC, and an NSERC DG and Discovery Accelerator Grants. Resources used in preparing this research were provided, in part, by the Province of Ontario, the Government of Canada through CIFAR, and companies sponsoring the Vector Institute https://vectorinstitute.ai/partners/. Additional hardware support was provided by John R. Evans Leaders Fund CFI grant and Compute Canada under the Resource Allocation Competition award. Finally, we would like to sincerely thank Muchen Li for valuable feedback and discussions.

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

# A  APPENDIX

## A.1  EFFECT OF PATCH SIZE

As we discuss in the main paper (Section 3.2), the control (mainly reduction) over the number of autoregressive steps can be achieved by simply varying the patch size $P$ in the `patch-random` model. The result of this on CIFAR10 are illustrated in Table 9. It can clearly be seen that a

|  | $1 \times 1$ | $2 \times 2$ | $4 \times 4$ | $8 \times 8$ | $16 \times 16$ |
|---|---|---|---|---|---|
| **LIN**(↑) | 59.79 | 69.63 | **75.53** | 75.34 | 60.77 |
| **FT**(↑) | 79.70 | 87.18 | **87.52** | 83.10 | 69.23 |

Table 9: **Patch-random** tokenization as a function of $P$ on CIFAR10.

different patch size $P$ does not lead to improved representation for a segment-free `patch-random` prediction task. The segment formation, on the other hand, as we show in the paper, does substantially improve the performance.

## A.2  BLOB SEGMENTS.

We define blob segments as irregular elliptical segments defined by a sampled Mixture of Gaussians. To obtain $K$ random blobs for a given image, we first sample $K$ Gaussians with means sampled from $[\mu_k^{(x)}, \mu_k^{(y)}] \sim \mathcal{U}(-1.75, 1.75)$ and standard deviations from $[\sigma_k^{(x)}, \sigma_k^{(y)}] \sim \mathcal{U}(0.5, 1)$, where $\mathcal{U}$ is a uniform distribution. Then we simply assign each token $\mathbf{x}_i$ which is at a normalized position $(x_i, y_i)$ in the range of $[-2, 2]$ (*i.e.*, leftmost top token is at (-2,-2), rightmost bottom token is at (2,2)). The assignment is done as follows:

$$S(\mathbf{x}_i) = \arg\max_k \mathcal{N}\left(\left[\begin{array}{c} x_i \\ y_i \end{array}\right] \Big| \left[\begin{array}{c} \mu_k^{(x)} \\ \mu_k^{(y)} \end{array}\right], \left[\begin{array}{cc} \sigma_k^{(x)} & 0 \\ 0 & \sigma_k^{(y)} \end{array}\right]^2\right). \tag{1}$$

$S$ is a function that maps tokens to segments. The sampling for both square and blob is only used during segment predictive training and is disabled during evaluation. The computation cost for sampling is, comparatively, negligible. Note that beyond the shape, blob segments allow for variability in size squares do not.

## A.3  TRAINABLE SKIP CONNECTIONS

We define a transformer design, with learnable skip connections, that we leverage for our main experiments in Section 4.2 of the main paper. Here we provide additional details and evaluation of that design which is illustrated in Figure 5 (right).

A transformer with $L_{enc}$ encoder layers and $L_{dec}$ decoder layers processes input $\mathbf{X}$ into $L_{enc}$ hidden representations $\mathbf{H}_{enc}^l = (\mathbf{h}_1^l, ..., \mathbf{h}_n^l)$. In traditional masked transformer, decoder memory is set to $\mathbf{Z}^l = \mathbf{H}_{enc}^{L_{enc}}$ for each layer $l$ of the decoder. Instead, we introduce a linear attention layer that allows each decoder layer to attend over encoding hidden representations. In other words, we learn a linear layer with parameters $\mathbf{W} \in \mathcal{R}^{L_{enc} \times L_{dec}}$, such that: $\mathbf{Z}^l = \sum_{k=1}^{L_{enc}} \mathbf{W}_{l,k} \mathbf{H}_{enc}^k$. Note, the linearly combined memory cells are conditioned on, and different, for each individual decoder layer.

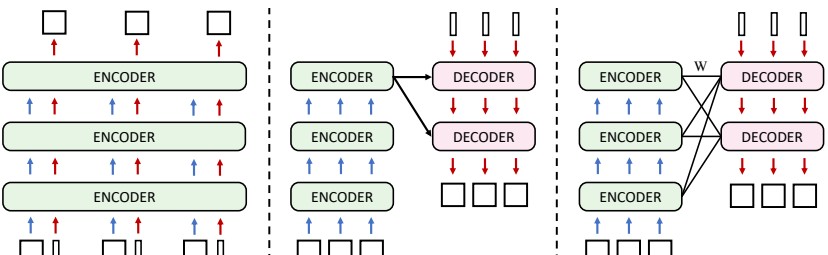

Figure 5: **Candidate architectures for autoregressive segment prediction:** Left: Two-stream Transformer (Yang et al., 2019). Middle: Masked Transformer. Right: proposed Masked Transformer with Trainable Skip Connections.

|  | CIFAR10 | | | | ImageNet100 | | | |
|---|---|---|---|---|---|---|---|---|
|  | Enc | Dec | LIN (↑) | FT (↑) | Enc | Dec | LIN (↑) | FT (↑) |
| Two-stream | 7M | 0M | 84.4 | 91.5 | 30M | 0M | 62.4 | 84.6 |
| Transformer | 5M | 2M | 88.0 | 93.6 | 21M | 9M | 65.8 | 85.9 |
| Transformer-skip | 5M | 2M | **89.5** | **94.4** | 21M | 9M | **70.7** | **87.3** |

Table 10: **Performance of Architectures for RandSAC**. See text for details.

To evaluate the effectiveness of the proposed Masked Transformer and trainable skip connection layer for segment prediction, we compare three architectures:

**Two-stream Transformer.** This design was proposed in (Yang et al., 2019) for permutation-based language modeling. It enables randomized target predictions by leveraging a two-stream attention layer: the *content* stream encodes the full contextual information, and the *query* stream, which only has access to the previous content, is designed to make current predictions. We apply this architecture for our segment prediction by setting the *content mask* with our "source mask" and *query mask* with our "memory mask". Model weights for both content stream and query stream are shared (see Figure 5 (left) for illustration of design).

**Masked Transformer.** For masked transformer we utilize architecture described in Section 4.1 and illustrated in Figure 4; also in Figure 5 (middle). Compared with the Two-stream Transformer above, this design enables communication among jointly predicted tokens within a segment. Also, compared with Two-stream Transformer, weights for encoding and decoding the segment content are decoupled in the Masked Transformer.

**Masked Transformer with Trainable Skip Connections.** A Masked Transformer only decodes based on the (last layer) encoder output. A trainable skip connection layer we introduce dynamically allocates memory assignments between intermediate layers of transformer-encoder-decoder (see Figure 5 (right)). As can be seen from the results in Table 10, this variant does outperform the two competitors on both CIFAR10 and ImageNet100 datasets. Compared with Masked Transformer, the additional computation cost introduced by trainable linear layer is almost negligible (Table 11).

|  | Two-stream | Transformer | Transformer-skip |
|---|---|---|---|
| ViT-T | 499.60 M | 483.82 M | 484.26 M |
| ViT-S | 10.94 G | 6.73 G | 6.77 G |
| ViT-B | 35.74 G | 21.25 G | 21.49 G |

Table 11: **FLOPs of Three Candidate Architectures.**

# B   EXPERIMENTS

## B.1   SEMANTIC SEGMENTATION ON ADE20K.

We take our pretrained backbone as initialization and end-to-end fine-tune with UpperNet framework on ADE20k to evaluate the performance of our pretrained model on downstream task, semantic segmentation. We follow the same setting of BeiT (Bao et al., 2022). We compare our pre-training with DeiT (Touvron et al., 2021), MoCo (Chen et al., 2021b), DINO (Caron et al., 2021), BeiT (Bao et al., 2022), MAE (He et al., 2021) in Table 12. Our pre-training outperform DeiT, MoCo, DINO, BeiT, MAE by 1.5, 1.3, 1.3, 2.0 and 0.4, respectively.

## B.2   OBJECT DETECTION ON COCO

We take our pretrained model as initialization and finetune with Mask RCNN on COCO. To adapt the the four stages designs with strides of 4, 8, 16, 32 of FPN backbone in Mask RCNN, we evenly divide all 12 Transformer blocks into 4 subsets and apply convolutions to upsample or downsample the intermediate feature maps for producing same scales as the requirement of FPN backbone. The results are reported in Table 13. Our pre-training outperform DeiT (Touvron et al., 2021), MoCo (Chen et al., 2021b), DINO (Caron et al., 2021), BeiT (Bao et al., 2022), and MAE (He et al., 2021) by 3.0, 3.0, 4.1, 1.1, 0.6 on $AP^{bbox}$ and 2.1, 2.3, 3.5, 0.6, 0.1 on $AP^{mask}$.

| Method | Crops | Super. | Self-super. | mIoU |
|---|---|---|---|---|
| DeiT (Touvron et al., 2021) | 1 | ✓ | ✗ | 47.0 |
| MoCo v3 (Chen et al., 2021b) | 2 | ✗ | ✓ | 47.2 |
| DINO (Caron et al., 2021) | 2+10 | ✗ | ✓ | 47.2 |
| BEiT (Bao et al., 2022) | 1 | ✗ | ✓ | 46.5 |
| MAE | 1 | ✗ | ✓ | 48.1 |
| RandSAC-Square (K=9) | 1 | ✗ | ✓ | **48.3** |
| RandSAC-Square (K=16→4) | 1 | ✗ | ✓ | **48.5** |

Table 12: **Semantic Segmentation on ADE20K**

| Method | Pre-Epochs | $AP^{bbox}$ | $AP^{mask}$ |
|---|---|---|---|
| DeiT (Touvron et al., 2021) | 300 | 47.9 | 42.9 |
| MoCo-v3 (Chen et al., 2021b) | 300 | 47.9 | 42.7 |
| DINO (Caron et al., 2021) | 300 | 46.8 | 41.5 |
| BEiT (Bao et al., 2022) | 800 | 49.8 | 44.4 |
| MAE (He et al., 2021) | 1600 | 50.3 | 44.9 |
| RandSAC-Square (K=16→4) | 1600 | **50.9** | **45.0** |

Table 13: **Object Detection on COCO**

### B.3 Visualization of Reconstruction on ImageNet-1k Validation Set

We visualize for both RandSAC-Blob and RandSAC-Square reconstruction results below.

### B.4 Implementation Details

We describe implementation details omitted from the main paper due to space limitations here.

**Implementation Details.** We adopt minimal data augmentation strategy following (He et al., 2021): resize cropping with scale range of $[0.2, 1.0]$ and aspect ratio is sampled within range $\left[\frac{3}{4}, \frac{4}{3}\right]$, followed by a $50\%$ chance random horizontal flipping. We do *not* use color jittering, path dropping, or gradient clip in pretraining. We use AdamW as optimizer and pretrain RandSAC for 1600 epochs. We use a linear *lr* scaling rule (Goyal et al., 2017) that scales the *base_lr* by *batchsize*/256. The *lr* is scheduled to warm-up from 0 to *base_lr*, then decayed following a cosine-decay rule (Loshchilov & Hutter, 2016). For both benchmarks, we use the *normalized pixel loss* introduced from (He et al., 2021) as our patch regression target. Our loss function computes the mean squared error (MSE) between the patch-normalized reconstruction and original image pixels.

### B.5 Evaluation protocols

**Linear Probing.** Note that the dimension of the feature that the classifier is trained on, may influence the eventual accuracy readout (Caron et al., 2021). A longer feature vector is likely to produce a better linear result. Prior works such as (Caron et al., 2021) concatenate the feature vectors from the last 4 ViT blocks and (Chen et al., 2020a) use feature vectors up to 15360 dimensions for evaluation. We, however, use only the last encoder averaged feature output following (He et al., 2021; Chen et al., 2021b) (*i.e.*, 384 dimensions for ViT-S and 768 dimensions for ViT-B). The linear classifier is trained for 90 epochs.

## C Details for Section 3

### C.1 Pre-training settings for CIFAR10 and ImageNet100

The following is the experiment configurations for CIFAR10 and ImageNet100 from Section 3 of the main paper, including Table 9 and 10 in Appendix. Details for end-to-end fine-tuning and linear

probing are the same with ImageNet-1K. For Table 10, both CIFAR10 and ImageNet100 experiments are trained using a "Blob" RandSAC model using hierarchy 11 →5.

### C.1.1 CIFAR10 Experiments.

The default setting is illustrated in Table 14. We pre-train ViT-Tiny encoder on CIFAR10. The ViT-Tiny has 12 layers. Each layer has 192 dimensions and 3 self-attention heads. We chose patch size $4 \times 4$ and split the $32 \times 32$ images into $8 \times 8$ tokens. For segment decoding, we use 3 transformer decoder layers following the same configuration for the encoder.

| config | value |
|---|---|
| optimizer | AdamW (Loshchilov & Hutter, 2019) |
| base_lr | 0.001 |
| weight decay | 0.05 |
| $\beta_1, \beta_2$ | 0.9, 0.999 (Carion et al., 2020) |
| batch size | 512 |
| learning rate schedule | cosine decay (Loshchilov & Hutter, 2016) |
| warmup epochs | 10 |
| training epochs | 800 (Table 1-6, 9) 1600 (Table 10) |
| augmentation | RandomResizedCrop |
| norm_pixel_loss | False (Table 1-6, 9) True (He et al., 2021) (Table 10) |

Table 14: **CIFAR10 Pre-training setting.**

### C.1.2 ImageNet100 Experiments.

Experiments that involve ImageNet100 are Table 2, 4, 6 and 10. We pre-train ViT-Small encoder on ImageNet100 (Tian et al., 2020). The ViT-Small backbone has 12 layers. Each layer has 384 dimensions and 6 self-attention heads. We chose patch size $16 \times 16$ following (Dosovitskiy et al., 2021) and split the $224 \times 224$ images into $14 \times 14$ tokens. For segment decoding, we use a 4 layer transformer decoder and double the attention heads while keeping all other configurations the same as the ViT-Small encoder.

| config | value |
|---|---|
| optimizer | AdamW (Loshchilov & Hutter, 2019) |
| base_lr | 1.5e-4 (He et al., 2021) |
| weight decay | 0.05 |
| $\beta_1, \beta_2$ | 0.9, 0.95 (Chen et al., 2020a) |
| batch size | 4096 |
| learning rate schedule | cosine decay (Loshchilov & Hutter, 2016) |
| warmup epochs | 40 |
| training epochs | 800 |
| augmentation | RandomResizedCrop |
| norm_pixel_loss | True |

Table 15: **ImageNet100 Pre-training setting.**

## D    Details for Section 4

### D.1    Low-data Pre-training Setting

We pre-train ViT-Small on **CIFAR10** and **CIFAR100** (Krizhevsky, 2009). Both datasets are small-scale image datasets containing 60000 $32 \times 32$ images that belong to 10 and 100 categories, respectively. The ViT-Small has 12 layers. Each layer has 384 dimensions and 6 self-attention heads. We chose patch size $4 \times 4$ and split the $32 \times 32$ images into $8 \times 8$ tokens. For segment decoding, we use a 6 transformer decoder layer. The attention-head and feature dimensions of the decoder are the

same as the encoder. We also set the decoder for MAE (He et al., 2021) to have the same depth, attention head, and dimension as ours.

| config | value |
|---|---|
| optimizer | AdamW (Loshchilov & Hutter, 2019) |
| base_lr | 0.001 |
| weight decay | 0.05 |
| $\beta_1,\beta_2$ | 0.9, 0.999 (Carion et al., 2020) |
| batch size | 512 |
| learning rate schedule | cosine decay (Loshchilov & Hutter, 2016) |
| warmup epochs | 10 |
| training epochs | 1600 |
| augmentation | RandomResizedCrop |
| norm_pixel_loss | True (He et al., 2021) |

Table 16: **Low Data Pre-training setting.**

## D.2 IMAGENET-1K PRE-TRAINING SETTING

We resize the images to $192 \times 192$ during pretraining and set patch size $P$ to be 16. We pretrain square-RandSAC with hierarchy 16 →4 using ViT-Base (Dosovitskiy et al., 2021) on ImageNet-1K following (Bao et al., 2022; He et al., 2021). ViT-Base model has 12 blocks, with each block having dimension 768 and 12 heads. We chose an 8 layer decoder. The attention-head and feature dimensions of the decoder are the same as the encoder.

| config | value |
|---|---|
| optimizer | AdamW (Loshchilov & Hutter, 2019) |
| base_lr | 1.5e-4 (He et al., 2021) |
| weight decay | 0.05 |
| $\beta_1,\beta_2$ | 0.9, 0.95 (Chen et al., 2020a) |
| batch size | 4096 |
| learning rate schedule | cosine decay (Loshchilov & Hutter, 2016) |
| warmup epochs | 40 |
| training epochs | 1600 |
| augmentation | RandomResizedCrop |
| norm_pixel_loss | True |

Table 17: **ImageNet-1K pre-training setting.**

## D.3 EVALUATION CONFIGURATIONS

Different from ViT (Dosovitskiy et al., 2021), where an additional class token is required for classification, we directly use the averaged pooled feature out of the encoder for both fine-tuning and linear probing. The hyper-parameters for both end-to-end finetuning and linear probing from Table 18 and Table 19 are used for all experiments of this paper.

## E IMPLEMENTATION DETAILS OF SEMANTIC SEGMENTATION

We end-to-end fine-tune our pre-trained ViT encoder with UpperNet framework on ADE20k to evaluate the performance on downstream task, semantic segmentation. We follow the same setting of BeiT (Bao et al., 2022). We take AdamW as the optimizer and set the batch size to 16, the layer-wise decay rate to 0.65, the input resolution to $512 \times 512$, fine-tuning iterations are set to 160K steps. During evaluation, we do not take multi-scale testing strategy in our experiment.

| config | value |
| --- | --- |
| optimizer | AdamW |
| base_lr | 5e-4 |
| weight decay | 0.05 |
| $\beta_1,\beta_2$ | 0.9, 0.999 (Chen et al., 2020a) |
| layer-wise lr decay | 0.65 |
| batch size | 1024 |
| learning rate schedule | cosine decay (Loshchilov & Hutter, 2016) |
| warmup epochs | 5 |
| training epochs | 100 |
| augmentation | RandAug (9, 0.5) (Cubuk et al., 2020) |
| label smoothing (Szegedy et al., 2016) | 0.1 |
| mixup (Zhang et al., 2017) | 0.8 |
| cutmix (Yun et al., 2019) | 1.0 |
| drop path (Huang et al., 2016) | 0.1 |

Table 18: **End-to-end fine-tuning setting.**

| config | value |
| --- | --- |
| optimizer | LARS (You et al., 2017) |
| base_lr | 0.1 |
| weight decay | 0 |
| momentum | 0.9 |
| batch size | 16384 |
| learning rate schedule | cosine decay |
| warmup epochs | 10 |
| training epochs | 90 |
| augmentation | RandomResizedCrop |

Table 19: **Linear probing setting.**

## F  CODE AND REPRODUCIBILITY

We include an implementation of RandSAC-Square model using PyTorch. We will release the complete training/evaluation code and all pre-trained models upon acceptance of the paper.

```python
import torch
import torch.nn as nn
import torch.nn.functional as F
from einops.layers.torch import Rearrange
from einops import rearrange
from torch import Tensor
from typing import Optional

class Transformer_skip(nn.Transformer):
    def __init__(self, num_encoder_layers: int = 6, num_decoder_layers:
    int = 4, **kwargs):
        """Transformer with learnable skip connects between encoder and
    decoder."""
        super().__init__(num_encoder_layers=num_encoder_layers,
                         num_decoder_layers=num_decoder_layers,
                         norm_first=True, **kwargs)
        self.skip_connection = nn.Linear(
            num_encoder_layers, num_decoder_layers)

    def forward(self, src: Tensor, tgt: Tensor, src_mask: Optional[Tensor
    ] = None, tgt_mask: Optional[Tensor] = None,
                memory_mask: Optional[Tensor] = None) -> Tensor:

        # Forward encoder layers
```

```python
22          memory = []
23          for layer in self.encoder.layers:
24              src = layer(src, src_mask=src_mask)
25              memory.append(src)
26
27          memory = self.encoder.norm(torch.stack(memory))
28
29          # Dynamic memory assignment
30          memory = self.skip_connection(
31              memory.flatten(1).transpose(0, 1)
32          ).transpose(0, 1).view((-1, *memory[0].shape))
33
34          # Forward decoder layers
35          for i, layer in enumerate(self.decoder.layers):
36              tgt = layer(tgt, memory[i],
37                          tgt_mask=tgt_mask, memory_mask=memory_mask)
38
39          return self.decoder.norm(tgt)
40
41 class RandSAC(nn.Module):
42      def __init__(self, d_model, image_channel=3, image_size=192,
        patch_size=16, M=4, **transformer_kwargs):
43          super().__init__()
44          """
45          RandSAC implementation with square segments and flat
        serialization (no hierarchy).
46          """
47          grid_size = image_size // patch_size
48          patch_dim = patch_size * patch_size * image_channel
49
50          self.M = M
51          self.patchify = Rearrange(
52              'n c (h p1) (w p2) -> n h w (p1 p2 c)', p1=patch_size, p2=
        patch_size)
53          self.in_proj = nn.Linear(patch_dim, d_model)
54
55          self.transformer = Transformer_skip(
56              d_model=d_model, **transformer_kwargs)
57
58          self.out_proj = nn.Linear(d_model, patch_dim)
59          self.pos = nn.Parameter(torch.zeros(1, grid_size, grid_size,
        d_model))
60          torch.nn.init.normal_(self.pos, std=.02)
61
62          self.register_buffer(
63              'mask', torch.repeat_interleave(
64                  torch.repeat_interleave(
65                      nn.Transformer.generate_square_subsequent_mask(
66                          sz=grid_size**2 // M**2 - 1
67                      ),
68                      repeats=M**2, dim=0
69                  ),
70                  repeats=M**2, dim=1
71              )
72          )
73
74      def serialize(self, patches):
75          """Flat serialization"""
76          d1, d2 = patches.shape[-1], self.pos.shape[-1]
77          tokens = torch.cat(
78              [patches, self.pos.repeat(patches.shape[0], 1, 1, 1)], dim
        =-1)
79          seq = rearrange(
80              tokens, 'n (h m1) (w m2) d -> n (h w) m1 m2 d', m1=self.M, m2
        =self.M)
```

```
81      noise = torch.rand(*seq.shape[:2], device=seq.device)
82      ids_shuffle = torch.argsort(noise, dim=1)
83      seq = torch.gather(seq, dim=1, index=ids_shuffle.view(
84          *seq.shape[:2], 1, 1, 1).expand_as(seq))
85
86      return seq.flatten(1, 3).transpose(0, 1).split([d1, d2], dim=-1)
87
88  def forward(self, img, label=None):
89      """Forward RandSAC"""
90      patches = self.patchify(img)
91      patches, pos = self.serialize(patches)
92
93      seg_size = self.M**2
94      embedings = self.in_proj(patches)
95
96      dec_out = self.transformer(src=(embedings + pos)[:-seg_size], tgt
    =pos[seg_size:],
97                                 src_mask=self.mask, tgt_mask=self.mask
    , memory_mask=self.mask)
98
99      pixel_recon = self.out_proj(dec_out)
100
101     loss = F.mse_loss(pixel_recon, patches[seg_size:])
102
103     return loss
```

## G  VISUALIZATION OF TOKENIZATION AND SERIALIZATION

We visualize different tokenization and serialization schemes, discussed in Section 3 of the main paper, in the video file included as part of the supplemental materials.

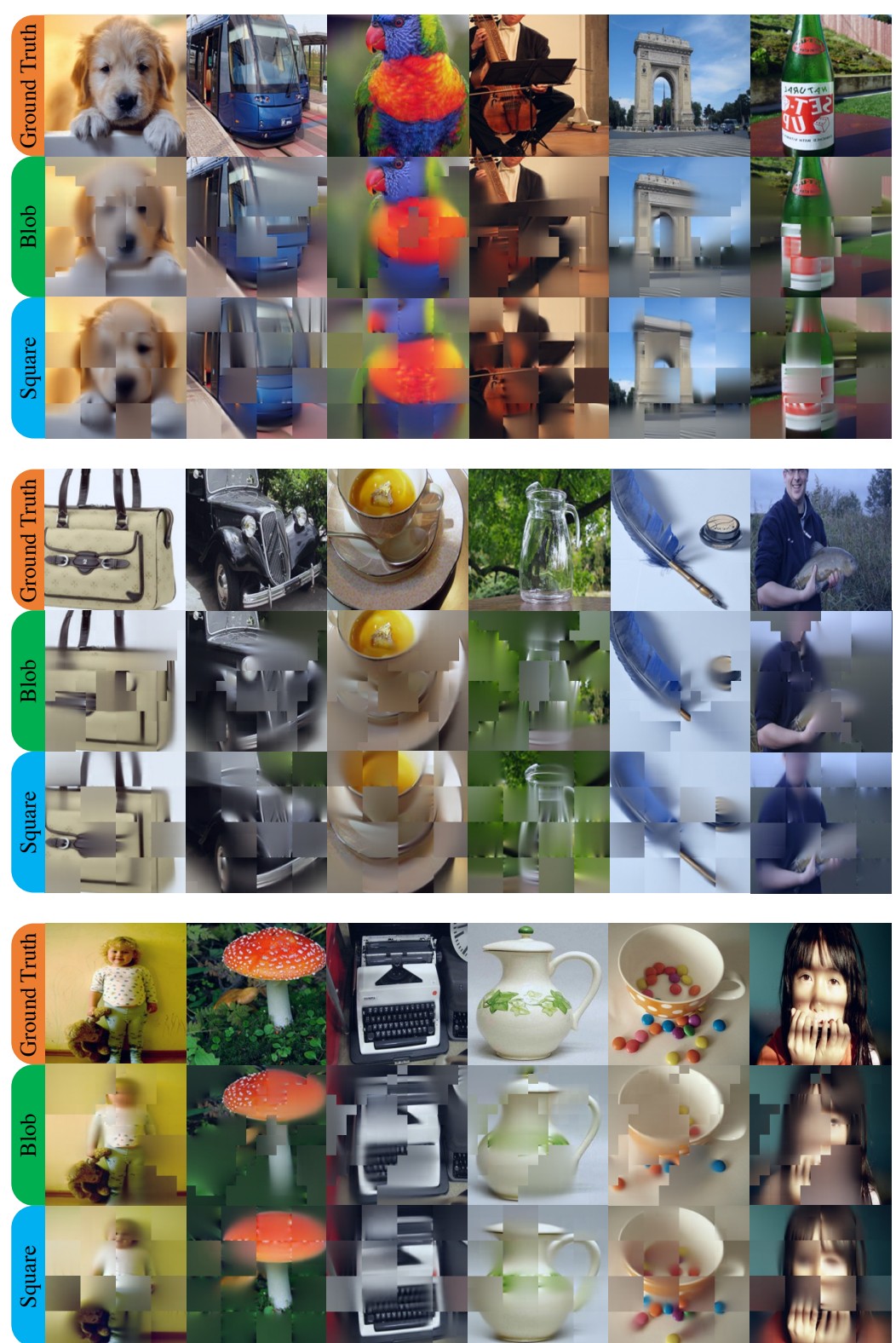

Figure 6: Visualization of image reconstruction from "Blob" and "Square" RandSAC.

