# OpenReview forum: "Self-supervision through Random Segments with Autoregressive Coding (RandSAC)"
_ICLR.cc/2023/Conference — ICLR 2023 poster_

### Official Review · Reviewer_pLid · 2022-10-22

**Confidence:** 3
**Correctness:** 3
**Technical Novelty And Significance:** 3
**Empirical Novelty And Significance:** 3
**Recommendation:** 6

**Clarity, Quality, Novelty And Reproducibility:**

+ Although the authors show their diverse vocabulary, however, this paper is not very easy to follow for the reviewer.
+ The content is not well organized. The reviewer suggests the authors to deliver the information in a way that how the proposed techniques are critical for the autoregressive mechanism. Currently, the proposed methods are not strongly aligned with autoregressive manner. Some of them can be also applied with autoencoding. Overall, the messages in this paper are somehow messy.
+ Since there are many details of the proposed framework, the reviewer is not sure whether it can be fully reproduced.


**Strength And Weaknesses:**

Issues about the autoregressive mechanism.
+ In NLP, the autoregressive pre-training paradigm might have more benefit for generative downstream tasks. The reviewer suggests the authors to elaborate more on this point and compare with the autoencoding counterpart. The reviewer believes that such discussion will make the paper standing out.
+ Currently, the authors simply demonstrate the effectiveness of this mechanism by showing the performance on downstream classification. The reviewer want to have more comprehensive impressions of the pretraining-finetuning gap of this mechanism. Could the authors conduct more downstream tasks such as detection and segmentation?
+ Since this paper focuses on the pre-training, the reviewer believes that using cifar dataset as the main example through the paper is somehow not convincing. As it is widely recognized that the scaling effect is very common in terms of the pre-training data, parameters, etc. As shown in the paper, the benefit of the proposed components become unsignificant when the data scale is increasing. To this end, the reviewer encourages the authors to use ImageNet as the main example and try to verify on models with larger capacities, such as the standard variant of MAE (i.e., ViT-Large).
***
Questions about the framework.
+ The reviewer finds that the decoder is heavier than that of MAE, since the dimensions of its hidden embeddings are enlarged to match with the encoder. Could the authors clarify how much the extra overhead this framework brings when compared with MAE?
+ Could the authors explain why the proposed methods are strongly aligned with the autoregressive modeling? The reviewer finds that the some of them, such as grouping the tokens as a segment or the skip connection in the transformer, can be applied to autoencoding modeling as well. The reviewer suggests the authors to show their strong connections with the autoregressive mechanism.
***
Other issues needed to be addressed.
+ In section 3, why the authors divide the tokenization strategy of BEiT and MAE into two categories. In fact, they share the same implementation to patchify images.
+ For table 7, there are also many contrastive learning based methods used on cifar. The authors can include them for comparison.
+ Current MAE can be regarded as predicting all the masked tokens sequentially at only one time. One the other hand, MAE can also be adjusted that reconstructing a partial ratio of masked tokens at every time sequentially. From the finetuning performance, improvement over MAE cannot be seen. Could the authors show the superiority of the proposed method against MAE?

**Summary Of The Paper:**

This paper focuses on autoregressive visual pre-training. The authors propose to spatially group tokens into a larger-resolution unit called segment, and then arrange them into a random order. After that, the authors sequentially predict the segments using transformer architecture with skip connection. The cifar dataset is used as the main benchmark to show the effectiveness of the proposed components.

**Summary Of The Review:**

+ Currently, the experimental results do not strongly prove the potential of the proposed framework in visual pre-training. The reviewer encourages the authors to conduct experiments with ImageNet and larger model since the scale effect is important for pre-training.
+ The authors should try to finetune their model on other downstream tasks beyond the same-dataset classification, in order to verify the gap between the proposed autoregressive pre-training and downstream transfer.
+ The authors should improve the writing and show how the proposed methods are strongly aligned with the autoregressive mechanism.

---

> ### Author Response · Authors · 2022-11-15
> **Response to Reviewer pLid (Part 1/2)**
>
>
>
>
> We appreciate all the constructive feedback from R3! We will address R3's questions and demonstrate the superiority of our method on ImageNet classification and downstream tasks in the following sections.
>
> ---
>
>
> ### Autoregressive pre-training paradigm might have more benefit for generative downstream tasks
>
> This is an interesting point. Indeed, autoregressive pre-training is likely to be very beneficial for generative downstream tasks. However, we believe that even for discriminative tasks it may also carry a benefit (a form of evidence for this is the strong downstream task performance we illustrate below). One technical difference between the masking and autoregressive pre-training we introduce is the distribution over complexity of the pre-training tasks. In MAE, and other masking methods, the complexity of the tasks is modulated by the masking ratio and, generally, remains fixed for the duration of pre-training. In our autoregressive pre-training, however, the conditioning window (number of variables/pixels model conditions on) varies across the sequence. This gives a distribution of complexities for pre-training tasks the variance of which can be controlled based on the size of the segments relative to the number of patches. We believe this affords our autoregressive model an added dimension of flexibility which is useful for pre-training.
>
> Further, one of our motivations for this work is to explore whether autoregressive pre-training strategies could indeed be as effective in vision as they have been in NLP for example. They have not been explored nearly as extensively as masking strategies in computer vision. Since this is a largely empirical question, we believe it variants some research and study. We hope our work is instrumental in shedding light in this respect.
>
> ---
>
> ### Could the authors show the superiority of the proposed method against MAE?
>
> The result in Table 8 of the original paper submission was based on an untuned learning rate of 2e-4. However, our method has superior performance when the learning rate is reduced to 1.5e-4, identical to that of MAE. The following table demonstrates our superiority against MAE.
>
> | Method     | Backbone    | Parameter   | Linear   | Fine-tune  |
> |   :---    |   :---:     |   :---:     | :---:    |      :---:     |
> |   MAE      |    ViT-B    |    86M      |   68.0   |   83.6     |
> | RandSAC-Square (K=9) | ViT-B | 86M       |  **72.3**    |   83.7 |
> | RandSAC-Square (K=16->4) | ViT-B | 86M   |  68.9    |  **83.9**  |
>
> ---
>
> ### Could the authors conduct more downstream tasks, such as detection and segmentation?
>
> Yes. While we have provided segmentation results in the original submission, we have since tuned the learning rate (see above) and have also conducted additional experiments on the detection downstream task. We will update both segmentation and detection downstream performance to the latest results shown below. As one can see, our results do beat MAE on both downstream tasks. In semantic segmentation by 0.4 mIoU and in detection by 0.6 AP; achieving SoTA performance.
>
> #### **Semantic Segmentation on ADE20K**
> We take our pretrained backbone as initialization and end-to-end fine-tune following the same settings of BeiT (Bao et al., 2022). We compare our pre-training with DeiT  (Touvron et al., 2021), MoCo (Chen et al., 2021b), DINO (Caron et al., 2021), BeiT (Bao et al., 2022), and MAE (He et al., 2022). Our pre-training outperforms DeiT, MoCo, DINO, BeiT, MAE by 1.5, 1.3, 1.3, 2.0, and 0.4, respectively.
>
> | Method | Supervised | Self-supervised | mIoU |
> | :---   |   :---:    |  :---:          | :---: |
> | DeiT   |     ✔       |                 | 47.0 |
> | MoCo v3 |           |          ✔       | 47.2 |
> | DINO   |            |          ✔       | 47.2 |
> | BEiT   |            |          ✔       | 46.5 |
> | MAE    |            |          ✔       | 48.1 |
> | RandSAC-Square (K=9) |         |   ✔   | **48.3** |
> | RandSAC-Square (K=16->4) |     |    ✔  | **48.5** |
>
>
> #### **Object Detection on COCO**
> We take our pretrained model as initialization and finetune it with Mask RCNN on COCO. Our pre-training outperforms DeiT  (Touvron et al., 2021), MoCo (Chen et al., 2021b), DINO (Caron et al., 2021), BEiT (Bao et al., 2022), MAE (He et al., 2022) by 3.0, 3.0, 4.1, 1.1, 0.6 on $AP^{bbox}$ and 2.1, 2.3, 3.5, 0.6, 0.1 on $AP^{mask}$. The results are reported below.
>
>
> | Method | Pre-Epochs | $AP^{bbox}$ | $AP^{mask}$ |
> | :---   |   :---:    |   :---:     |   :---:     |
> | DeiT   |   300      |    47.9     |   42.9      |
> | MoCo-v3 |  300      |    47.9     |   42.7      |
> | DINO   |   300      |    46.8     |   41.5      |
> | BEiT   |   800      |    49.8     |   44.4      |
> | MAE    |   1600     |    50.3     |   44.9      |
> | RandSAC-Square (K=16->4) | 1600 | **50.9** | **45.0**   |

---

> > ### Author Response · Authors · 2022-11-15
> > **Response to Reviewer pLid (Part 2/2)**
> >
> >
> > ### Size of the decoder compared to that of MAE
> >
> > Indeed, our decoder is wider than MAE's (768 dim vs. 512 dim). It is worth noting that in MAE, a wider decoder does not lead to better performance (See MAE Table 1b). Self-supervised representation learning aims to learn a pretrained backbone with transferable features. As long as it fits the computation budget, the decoder size wasn't our primary concern.
> >
> > ---
> >
> > ### The proposed methods, "segments" and "skip-connection", can also be used in Autoencoders.
> >
> > The idea of segments has already been explored extensively in Autoencoders, but it has not been explored in autoregressive models. The "segments" in our paper are essentially groups of spatially coherent tokens. Masked image modeling (MIM) / autoencoders adopt a similar idea called block-wise masking. Similar to segments in our paper, block-wise masking is to mask&inpaint images by blocks of tokens, tokens that are spatially connected. When they have been explored in MIM, the benefits vary. For example, in BEiT, block-wise masking is superior (See Table 4 of BEiT); in MAE and SimMIM, however, block-wise masking performs worse than random masking (See Table 1(f) of MAE and Table 1 of SimMIM).
> >
> >
> >
> > Skip-connection maps all intermediate layers in a transformer-encoder to the cross-attention layers in transformer-decoders dynamically. It increases the expressiveness of a transformer with only a few weights. In autoencoding modeling, they typically use transformer-encoder layers only. For example, in MAE, the decoder layers use the same layer in the encoder without the cross-attention layer usually seen in a transformer-decoder. It is unclear how skip-connection can be applied to autoencoding modeling approaches. In NLP and other computer vision tasks (e.g., DETR), where transformer autoencoders are used for sequence-to-sequence modeling, our skip-connection might be effective. Though it is beyond the scope of our paper, the design of learnable skip-connection is worth exploration in other transformer-based seq2seq architectures.
> >
> > ---
> >
> > ### In section 3, why do the authors divide the tokenization strategy of BEiT and MAE into two categories?
> > We apologize for the confusion. It was caused by a typo in citation. We will fix it in the revision.
> >
> > ---
> >
> > ### References
> > **DETR** Carion, Nicolas, et al. "End-to-end object detection with transformers." European conference on computer vision. 2020.
> >
> > **DINO**: Caron, Mathilde, et al. "Emerging properties in self-supervised vision transformers." Proceedings of the IEEE/CVF International Conference on Computer Vision. 2021.
> >
> > **DeiT**: Touvron, Hugo, et al. "Training data-efficient image transformers & distillation through attention." International Conference on Machine Learning. PMLR, 2021.
> >
> > **MoCo-v3** Chen, Xinlei, Saining Xie, and Kaiming He. "An empirical study of training self-supervised vision transformers." Proceedings of the IEEE/CVF International Conference on Computer Vision. 2021.
> >
> > **BEiT** Bao, Hangbo, Li Dong, and Furu Wei. "Beit: Bert pre-training of image transformers." International Conference on Learning Representations. ICLR, 2021.
> >
> > **MAE** He, Kaiming, et al. "Masked autoencoders are scalable vision learners." Proceedings of the IEEE/CVF Conference on Computer Vision and Pattern Recognition. 2022.
> >
> > **SimMIM** Xie, Zhenda, et al. "Simmim: A simple framework for masked image modeling." Proceedings of the IEEE/CVF Conference on Computer Vision and Pattern Recognition. 2022.

---

> > > ### Comment · Reviewer_pLid · 2022-12-11
> > > **Thanks for the responses**
> > >
> > > Thanks for the responses, which have addressed most of my concerns.

---

### Official Review · Reviewer_qvKL · 2022-10-24

**Confidence:** 4
**Correctness:** 4
**Technical Novelty And Significance:** 4
**Empirical Novelty And Significance:** 4
**Recommendation:** 8

**Clarity, Quality, Novelty And Reproducibility:**

The presented method is novel. I do not have comments regards the reproducibility as no source-codes are provided.

**Strength And Weaknesses:**

Strength:
1. Great paper writing with clear and easy understanding motivation.
2. Good empirical results and comprehensive ablations.



**Summary Of The Paper:**

This paper presents an interesting method continuing the previous auto-regressive self-supervised learning schema. Unlike the existing iGPT which operates on pixel-leve in a fixed order, RandSAC proposes to regress the image tokens in sequential (across grouped image segments) and parallel fashions (within segment). Surprisingly, with this simple method, the exhibited empirical results are optimistic and much efficient compared with iGPT.

**Summary Of The Review:**

The presented RandSAC builds upon the previous auto-regressive self-supervised visual representation fashion (i.e., iGPT). Compared with existing work, the proposed grouping and regressive fashion is simple but effective. The proposed hierachical grouping strategy brings obvious goodness: 1. allows various length learning on visual segments. 2. It largel improves the efficiency of pre-training.

---

> ### Author Response · Authors · 2022-11-15
> **Response to Reviewer qvKL**
>
> We thank R2 for the positive feedback!

---

### Official Review · Reviewer_FF3A · 2022-10-25

**Confidence:** 4
**Correctness:** 3
**Technical Novelty And Significance:** 3
**Empirical Novelty And Significance:** 3
**Recommendation:** 8

**Clarity, Quality, Novelty And Reproducibility:**

- (Clarity) The paper is clearly written and easy to follow.
- (Quality) The experiments were well organized and clearly addressed the research questions brought throughout the paper.
- (Novelty) I consider this model to be novel enough as it delivers new ways of grouping tokens and ordering the prediction sequence.
- (Reproducibility) Source code is not provided, but implementation details and the pseudo-code in the appendix provide reproducibility.

**Strength And Weaknesses:**

- (C1) There have not been many papers studying ordering or grouping methods in autoregressive methodologies, I would like to give a high score to find a way to do it well.
- (C2) I believe DINO [1] is not a contrastive learning method but instead bootstrapping or self-distillation-like method (e.g., BYOL [2]). We can find another stream of works, such as SwAV [3] and DeepCluster [4,5], which are not contrastive learning, masked image modeling, or bootstrapping (these methods are better classified as a clustering method). Thus, coarsely dividing SSL methods into contrastive and predictive categories is not appropriate.
- (C3) CIFAR10 and CIFAR100 experiments in table 7 are rather low-resolution pre-training than low-data pre-training. Although it is a worthy experiment, I think the authors can do an eye-to-eye low-data pre-training by limiting the number of examples for the ImageNet-1K dataset.
- (C4) I am confused about whether or not the learnable skip connection module is used for the experiments in section 5. Plus, is it okay for the learned matrix W to be trained without any constraints (e.g., \sum_{k=1}^{L_{enc}} W_{l, k} = 1)? If so, will the constrained version worsen the performance?
- (C5)
  - Typo: (Section 5) Suppl. Sec. D.3 -> B.1.
  - Typo: (Section 5) Supplemental Appendix B → D

[1] Caron, Mathilde, et al. "Emerging properties in self-supervised vision transformers." *Proceedings of the IEEE/CVF International Conference on Computer Vision*. 2021.
[2] Grill, Jean-Bastien, et al. "Bootstrap your own latent-a new approach to self-supervised learning." *Advances in neural information processing systems* 33 (2020): 21271-21284.
[3] Caron, Mathilde, et al. "Unsupervised learning of visual features by contrasting cluster assignments." *Advances in Neural Information Processing Systems* 33 (2020): 9912-9924.
[4] Caron, Mathilde, et al. "Unsupervised pre-training of image features on non-curated data." *Proceedings of the IEEE/CVF International Conference on Computer Vision*. 2019.
[5] Caron, Mathilde, et al. "Deep clustering for unsupervised learning of visual features." *Proceedings of the European conference on computer vision (ECCV)*. 2018.

**Summary Of The Paper:**

This paper delves deep into ordering and grouping methods for causal image modeling. Moving on from iGPT, which used quantized RGB tokens for its causal prediction, this paper lets the models predict the pixel values directly as MAE does. When it comes to ordering, the paper showed that random ordering could yield much better performance compared to conventional raster ordering. And for the grouping (segmentation) method, two approaches were square segments and blob segments, and both performed better than no segment baseline. Further, the paper proposed a learnable skip connection module connecting the transformer encoder and decoder layers.

**Summary Of The Review:**

It would be nice if some of my questions (see C2-C4) were answered, but I want to give this paper a high recommendation because its strengths far outweigh its weaknesses.

---

> ### Author Response · Authors · 2022-11-15
> **Response to Reviewer FF3A**
>
> We thank R1 for acknowledging our work as "clear", "well organized" and "novel enough". We also appreciate your valuable feedback on our paper. We will address R1's questions below.
>
> -----
>
> ### (C2) DINO is a bootstrapping or self-distillation-like method.
> We apologize for coarsness of our classification for SSL methods. Indeed, DeepCluster, SwAV, and DINO can all be considered clustering-based methods. We will reclassify them in our latest version of the paper.
>
> -----
>
> ### (C4) Questions regarding the skip-connection.
> Yes, the skip connections are used for experiments in Section 5. In our experiments, the constraint of having all incoming weights for the skip-connection summing to 1 does, in fact, worsen the performance, as can be seen from the results below. It causes no issues not normalizing the learned weights, as the model will implicitly learn the appropriate relative scaling of contribution from different encoder layers.
>
> |Method | Linear | Finetune |
> | :---  | :---:  | :---: |
> |skip-connection | 67.7 | 82.9 |
> |skip-connection w/ softmax | 66.9 | 82.9 |
>
> Experiment settings:
>
> |Backbone | ViT-B |
> |:---     | :---: |
> |Decoder layers | 8 |
> | Pretrain epochs | 300 |
> |Segment | Square |
> |Image size | 192 |
> |Hierarchy | 16->9|
> | Learning rate | 1.5e-4 |
> -----
> ### (C5) Typos
>
> Thank you, these have been fixed.

---

### Decision · Program_Chairs · 2023-01-20

**Decision:**

Accept: poster

**Justification For Why Not Higher Score:**

The results are interesting, but not a dramatic improvement on prior work.

**Justification For Why Not Lower Score:**

All reviewers recommended accepting the paper.

**Metareview: Summary, Strengths And Weaknesses:**

The paper proposes a generative method for self-supervised learning on images which divides images into tokens, then models tokens autoregressively. All reviewers appreciated the novelty of the approach, and were generally satisfied with the empirical results. Reviewer pLid raised many relevant issues in their initial review, including more discussion, better comparison with MAE, and more downstream tasks. The authors submitted an extensive response including new experimental results which were successful in mostly addressing these concerns. In the end all reviewers recommended accepting the paper. The authors are encouraged to take the suggestions of Review pLid into account when preparing the camera-ready version of the paper.

**Note From Pc:**

if the above contains the word "oral" or "spotlight" please see: "oral" presentation means -> notable-top-5% and "spotlight" means -> notable-top-25%. As stated in our emails, we are disassociating presentation type from AC recommendations